# Preparation of Apixaban Solid Dispersion for the Enhancement of Apixaban Solubility and Permeability

**DOI:** 10.3390/pharmaceutics15030907

**Published:** 2023-03-10

**Authors:** Juseung Lee, Jong-Ju Lee, Seungyeol Lee, Linh Dinh, Hangyu Oh, Sharif Md Abuzar, Jun-Hyun Ahn, Sung-Joo Hwang

**Affiliations:** 1College of Pharmacy, Yonsei University, 85 Songdogwahak-ro, Yeonsu-gu, Incheon 21983, Republic of Korea; 2Yonsei Institute of Pharmaceutical Sciences, Yonsei University, 85 Songdogwahak-ro, Yeonsu-gu, Incheon 21983, Republic of Korea

**Keywords:** apixaban, anticoagulation, solid dispersion, solubility enhancement, bioavailability enhancement

## Abstract

(1) Background: Solid dispersion (SD) can help increase the bioavailability of poorly water-soluble drugs. Meanwhile, apixaban (APX)—a new anticoagulation drug—has low water solubility (0.028 mg/mL) and low intestinal permeability (0.9 × 10^−6^ cm/s across Caco-2 colonic cells), thus resulting in a low oral bioavailability of <50%; (2) Methods: To solve the drawbacks of conventional APX products, a novel SD of APX in Soluplus^®^ was prepared, characterized by differential scanning calorimetry (DSC), powder X-ray diffraction (PXRD) and Fourier transform infrared (FTIR) spectroscopy techniques and evaluated for its solubility, intestinal permeability and pharmacokinetic performance. (3) Results: The crystallinity of the prepared APX SD was confirmed. The saturation solubility and apparent permeability coefficient increased 5.9 and 2.54 times compared to that of raw APX, respectively. After oral administration to the rats, the bioavailability of APX SD was improved by 2.31-fold compared to that of APX suspension (4) Conclusions: The present study introduced a new APX SD that potentially exhibits better solubility and permeability, thus increasing APX’s bioavailability.

## 1. Introduction

Apixaban (APX) (Figure 1) is a potent, new, reversible, direct and highly selective anticoagulation drug. By inhibiting Factor-Xa, APX prevents thrombin generation and thrombus development inhibiting coagulation induced by thrombin while having no direct effects on platelet aggregation [1]. The maximum anticoagulation effect of APX on pharmacodynamic parameters occurs at the time as APX plasma concentration peaks at 3–4 h after drug administration [2]. APX is absorbed primarily in the small intestine, extensively metabolized via CYP3A4/5 [3] and heavily affected by the P-glycoprotein efflux system [3,4] (intestinal permeability of only 0.9 × 10^−6^ cm/s across Caco-2 colonic cells) [4]; thereby, APX decreases progressively in the gastro-intestinal tract. APX exhibits a fairly low oral availability of <50% [3,5] due to its incomplete absorption in the intestine and multiple routes of elimination [3]. According to the Biopharmaceutical Classification System (BCS) because the therapeutic dose of APX (5 mg) is soluble in 250 mL of physiological buffer over the pH range of 1.2–6.8 at 37 °C [6,7], APX is classified as class III. But APX solubility of 0.028 mg/mL [8] is “practically insoluble” when referring to the USP/NF. The discrepancy between BCS and APX absolute physicochemical properties was because dose is considered in the classification of solubility [9]. There have been studies about the enhancement of APX solubility, ultimately to enhance its oral bioavailability [10,11,12]. Notably, solid dispersion (SD) has been employed in studies by Zhoye Y.Y and Asati et al.; APX SD with mannitol and several hydrophilic polymers were prepared showing improvement of APX in aqueous solution [11,12]. 

SD is a mixture of (an) active ingredient(s) in an inert hydrophilic carrier at solid state [13]. Successful dispersion of the drug in the carrier, at a molecular level, overcomes the intermolecular force between drug molecules and leads to the formation of the homogeneous phase [13,14] of the SD, which can help increase the solubility and dissolution rate of poorly water-soluble drugs with consequent improvement in bioavailability. There are many poorly soluble drugs with low dissolution rate, low solubility and poor oral availability, which remain challenging for formulation scientists; however, not many SD products are commercially available. SDs are normally soft, bulky and tacky, thus making them difficult to fill in capsules and/or compress into a tablet.

Currently, APX is commercially available as 2.5 mg and 5 mg doses in the form of an oral film-coated immediate release tablet, which is approved for the treatment of deep vein thrombosis, pulmonary embolism and for the reduction of recurrent of deep vein thrombosis and pulmonary embolism following initial treatment for venous thromboembolism [15]. However, the major side effects of APX are gastrointestinal bleeding, nausea and abdominal discomfort. Because of the frequent side effects and APX interactions with other drugs and food, taking APX requires strictly routine monitoring [16]. The goal is to adequately formulate APX into a drug product that not only ensures the pharmacological therapeutic effect but also makes patients feel more comfortable. 

In this study, we prepared several novel APX SDs. These APX SDs were characterized and evaluated for their solubility, permeability and oral bioavailability in Sprague-Dawley rats. 

## 2. Materials and Methods

### 2.1. Materials

Apixaban (APX) was kindly provided by MSN Laboratories (Telangana, India). Rivaroxaban was obtained from GLPharmTech (Gyeonggi-do, Republic of Korea). Soluplus^®^ was purchased from BASF SE (Ludwigshafen, Germany). Methanol (gradient grade = 99.9), water (gradient grade = 99.9) and acetonitrile (ACN) (gradient grade = 99.9) were purchased from Fisher Scientific (Pittsburgh, PA, USA). Acetone, 2-propanol, ethyl acetate, methylene chloride, chloroform, formic acid, potassium phosphate monobasic, potassium phosphate dibasic, sodium phosphate monobasic, sodium phosphate dibasic and poly (vinyl alcohol) (PVA) were purchased from Samchun Chemicals (Gyeonggi-do, Republic of Korea). Eudragit^®^ L100, Eudragit^®^ S100, and Eudragit^®^ E PO were purchased from Evonik Industries (Essen, Germany). Kollidon^®^ VA 64, Kollidon^®^, Soluplus^®^, Kolliphor^®^ P 407, Kolliphor^®^ P 188 and Kolliphor^®^ TPGS were purchased from BASF (Ludwigshafen, Germany). Sodium lauryl sulfate was purchased from Duksan Reagents (Gyeonggi-do, Republic of Korea). Krebs-Ringer bicarbonate (KBR) buffer and dimethyl sulfoxide (DMSO) were purchased from Sigma-Aldrich (St. Louis, MO, USA). Purified water used was from the Milli-Q system (Milli–Q reference, Millipore^®^, Molsheim, France).

Male Sprague-Dawley rats, 7 weeks old, weighing approximately 250 g were purchased from Samtako Bio (Osan, Republic of Korea), and used for animal studies. The animals were housed in a semi-specific pathogen-free facility using standard cages at 21 ± 1 °C and 65% relative humidity with a 10 h light/14 h dark cycle. The rats were given standard chow and water ad libitum for the duration of the study and an extra 2 weeks prior to the study to adapt to the environment. The rats were allowed free access to food and water and were fasted for 24 h before the experiments.

### 2.2. Methods

#### 2.2.1. APX Solubility Study in Organic Solvents

To determine the specific organic solvent used for preparing APX solid dispersion (SD), the saturation solubility of APX in various organic solvents such as methanol, ethanol, acetonitrile, acetone, 2-propanol, ethyl acetate, methylene chloride and chloroform was screened. An excess amount of APX was put into each solvent and sonicated for 30 min, followed by putting the samples in a CRT-350 Rotator (Lab Companion, Daejeon, Republic of Korea) at 30 rpm for 72 h. Samples then were centrifuged at 10,000 rpm for 15 min and filtered using 0.45 µm polytetrafluoroethylene (PTFE) syringe filters. After suitable dilution, aliquots were withdrawn and the concentration of APX was determined by high-performance liquid chromatography (HPLC).

#### 2.2.2. APX Solubility Study with Hydrophilic Carriers 

Various polymers and surfactants were dissolved in 5 mL of distilled water at concentrations of 0.1%, 0.5%, 1% (*w*/*v*), and an excess amount of APX was added into each polymer–surfactant solution. After sonication for 30 min, the samples were placed in a shaking water bath at 50 rpm, 37 °C for 72 h. The samples were then centrifuged at 10,000 rpm for 15 min and filtered using 0.45 µm PTFE syringe filters; aliquots were withdrawn and the concentration of APX was determined by HPLC.

#### 2.2.3. Preparation of SDs

A preliminary experiment was conducted by the kneading method to choose the appropriate hydrophilic carrier among the studied polymers (Kollidon^®^, Kolliphor^®^ P 188, Kolliphor^®^ P 407, Kollidon^®^ VA 64, Soluplus^®^ and Kolliphor^®^ TPGS) and the drug-to-hydrophilic carrier weight ratio was 1:5. Then, 50 mg of APX and 250 mg of a hydrophilic carrier were put into a mortar and 500 µL of acetone was added. By kneading gently using a pestle for 15 min, APX was dispersed into the hydrophilic carrier. The suspension-like dispersed system was left to dry in an oven (Lab Companion, Daejeon, Republic of Korea) at 60 °C for 24 h. Dried powder was collected and stored at room temperature.

SDs of APX with hydrophilic carriers chosen from the preliminary experiment were prepared by the solvent evaporation method using a rotary evaporator (Eyela N-1110, Tokyo, Japan). APX and a hydrophilic carrier at different weight ratios (1:3, 1:5 and 1:10) were dissolved in acetone; the APX concentration in acetone was prepared at 1 mg/mL. The mixtures were sonicated until both APX and the hydrophilic carrier were fully dissolved, and then the solvent (acetone) was removed after 2 h of putting it under vacuum. The precipitated SD samples were gently collected and placed in a desiccator for additional drying. Obtained SDs were carefully weighed, sieved (60 mesh, 250 µm) to reach fine powder form and stored at room temperature. To confirm the actual content of APX in the prepared SDs, an amount of an SD sample was dissolved in methanol and then aliquots were subjected to HPLC to determine the concentration of APX. 

#### 2.2.4. Differential Scanning Calorimetry (DSC) Analysis

DSC analysis was conducted by AUTO-DSC Q2000 (TA instruments, New Castle, DE, USA) to monitor the calorimetric responses of APX and hydrophilic carrier in the SD system. All samples were accurately weighed to 3–5 mg in an aluminum pan, then were sealed in standard aluminum pans with lids. A blank aluminum pan was used as a reference. Samples were scanned from −10 to 300 °C at an increasing heating rate of 10 °C/min. A nitrogen flow rate of 40 mL/min was used. 

#### 2.2.5. Fourier Transform Infrared Spectroscopy (FT-IR) Analysis

FT-IR spectra of the samples were collected by using a Cary 630 FTIR spectrometer (Agilent, CA, USA) equipped with an ATR accessory (ZnSe crystal). Spectra were collected over the scan range of 650–4000 cm^−1^ with a resolution of 4 cm^−1^. Thirty-two scans were collected.

#### 2.2.6. Powder X-ray Diffraction (PXRD) Analysis

PXRD patterns were measured by Rigaku Ultima IV X-ray diffraction system (Rigaku, Tokyo, Japan) in the θ/2θ scan mode with Cu K-α radiation at 40 kV and 40 mA. The sample was loaded in a small disc-like container and its surface was carefully flattened. Diffraction patterns were collected in the 2θ range after a run in the range of 5 to 50° with 0.02° step size at a rate of 2°/min.

#### 2.2.7. Disc Intrinsic Dissolution Rate (DIDR) Study

The DIDRs of APX and APX SDs were measured by using a stationary disk method [17]. DIDR is not an absolute measurement of the drug solubility but a rate measurement phenomenon; the surface area from which dissolution takes place must be recognized and kept constant. This was achieved by die compressing samples with a cavity of determined diameter to produce a disk of known surface area. Only one side of the compressed powder is exposed to the dissolution medium by the die cavity. The disk was prepared with sufficient compressing force in order not to disintegrate in the dissolution medium. The DIDR (*j*) was calculated as
(1)$$j=V\times \frac{dC}{dt}\times \frac{1}{A}$$
where *j* is the DIDR, *V* is the volume of the dissolution medium, *C* is the concentration, *A* is the area of the drug disk, and *t* is the time [9,17]. 

Briefly, 100 mg of APX or APX SD were precisely weighed and then put into the die cavity (8 mm in diameter) and compressed by a PerkinElmer hydraulic press (PerkinElmer Corp., Norwalk, CT, USA) for 0.5 min at 1000 psi to obtain a disk with a surface area of approximately 0.5 cm^2^_._ The disk was placed in 500 mL of dissolution medium (50 mM sodium phosphate buffer with 0.05% sodium lauryl sulfate, pH 6.8) at 37 ± 0.5 °C. The stirring paddle was positioned at an appropriate distance of 1.0 in. (2.54 cm) above the disk surface and rotated at 75 rpm. At predetermined time intervals, aliquots of 1 mL were withdrawn and compensated with dissolution medium right after sampling. After filtering using 0.45 µm PTFE syringe filters, the concentration of APX was measured by HPLC.

#### 2.2.8. Powder Dissolution Test

The powder dissolution test was conducted using USP Apparatus 2 (paddle) method, DST-810 dissolution tester (Labfine. Inc., Republic of Korea). Because APX and the prepared APX SD are not sunk into the dissolution medium, a certain weight of sample equivalent to 5 mg APX was capsulated in a capsule (size 2), put inside a stainless-steel sinker (2.4 × 1.2 cm) and then put into the vessels to prevent the capsules from floating [18] on the surface of the dissolution medium (500 mL of 50 mM sodium phosphate buffer with 0.05% sodium lauryl sulfate, pH 6.8). The media were preheated to 37 ± 0.5 °C and stirred at 75 rpm. At predetermined time intervals (5, 10, 20, 30, 45, 60, 90, 120 and 180 min), aliquots of 1 mL were withdrawn and compensated with dissolution medium right after sampling. After filtering using 0.45 µm PTFE syringe filters, the concentration of APX was measured by HPLC.

#### 2.2.9. Ex Vivo Permeation Test

The increased intestinal permeability from apical to basolateral side of APX SDs compared to that of APX was examined by modified Ussing chamber technique [19] using rat jejunum. Male Sprague-Dawley rats were anesthetized by isoflurane inhalation. After anesthetization, the rats were sacrificed, and their small intestines were isolated. Each jejunum was cleaned with saline solution and cut into segments. Two pieces of jejunum (0.5 cm^2^) were placed on an acrylic plate and fixed in place of the Ussing chamber system (Easymount, Arlington, TX, USA). Then, 2 mL of KBR buffer (pH 7.4, composed by 10 mM D-glucose, 0.5 mM MgCl_2_, 4.6 mM KCl, 120 mM NaCl, 0.7 mM Na_2_HPO_4_ and 15 mM NaHCO_3_) with 5% DMSO solution was added to both the apical and basolateral sides and incubated with carbogen (O_2_/CO_2_ 95%/5%) gas at 37 ± 0.5 °C for 30 min. Drug donor solutions were prepared by dissolving APX or APX SD in KBR with 5% DMSO solution to obtain APX concentration equivalent to 100 μg/mL. A total of 2 mL of drug solution was added to the apical side; therefore, the final concentration of APX was 50 μg/mL. An equivalent volume of KBR with 5% DMSO was added to the basolateral side. At predetermined time intervals, aliquots of 500 μL were withdrawn from the basolateral side and immediately replaced with 500 μL of medium. The aliquots were then subjected to HPLC analysis. The apparent permeability coefficient was calculated as
(2)$$P_{app}=\frac{Q}{C\times A\times t\;}$$
where *Q* is the total amount of APX that permeated to the basolateral side, *C* is the initial drug concentration in the apical side, *A* is the diffusion area of the Ussing chamber, and *t* is the time [20].

#### 2.2.10. HPLC Analysis

The HPLC APX assay was performed by using the Agilent 1260 Infinity LC system (Agilent Technologies, Santa Clara, CA, USA). The samples were measured at 270 nm by using an HPLC-UV spectrometer (Agilent 1290 infinity) with a C18 column (InfinityLab ZORBAX 300SB, 120 Å pore size, 5 mm, 4.6 mm inside diameter ×250 mm, Agilent Technologies, Santa Clara, CA, USA). The mobile phase comprised a 60:40 mixture of potassium phosphate buffer 0.05 M, pH 3.5) and ACN, pumped (Model 1260 Quat Pump VL) at a rate of 1 mL/min. The samples were diluted as required before the injection of 20 µL using an auto sampler (Model 1260 ALS). The auto sampler temperature and column temperature were set at 25 °C.

#### 2.2.11. In Vivo Pharmacokinetic Study 

APX and APX SDs were dispersed in distilled water and 2 mg/mL was administered orally to male Sprague-Dawley rats aged 7 weeks using sonde. The rats were randomly divided into control and treatment groups (n = 5). Control groups received APX per oral (PO); treatment groups received APX SD PO. After the administration, rats were anesthetized by isoflurane inhalation and 0.1 mL of blood was collected from the tail vein at predetermined time points. Blood samples were centrifuged at 13,000 rpm for 10 min at 4 °C. Supernatant plasma was obtained and stored for analysis. The concentration of APX in rat plasma was analyzed by liquid chromatography–tandem mass spectrometry (LC–MS/MS) with a validated bioanalytical method. 

#### 2.2.12. LC-MS/MS Analysis 

Bioanalytical method validation was conducted to attain reliable data from the plasma sample. In this study, full validation was conducted within the acceptance criteria of the validation guidelines from US FDA 2018. 

Chromatographic separation was performed by Agilent 1290 Infinity LC system (Agilent Technologies, Santa Clara, CA, USA). An XBridge C18 column (2.1 mm × 30 mm, 3.5 μm; Waters Corporation, Milford, MA, USA) was used. The binary mobile phase composed of 60% water (0.1% formic acid) and 40% ACN (0.1% formic acid) was pumped at a total flow rate of 1 mL/min. The auto sampler temperature and column temperature were set at 4 °C and 25 °C, respectively. The injection volume was 3 μL. MS/MS analysis was performed by a 6490 Triple Quadruple tandem mass spectrometer (Agilent Technologies, Santa Clara, CA, USA) equipped with an electrospray ionization source operating in the positive-ion mode. The multiple reaction monitoring acquisition method was used to detect the analytes with m/z 460.2–443.02 for APX and 436.11–144.96 for rivaroxaban (internal standard) in collision energy 35 V and 30 V, respectively. The nozzle voltage was 1500 V and nebulizer pressure was 20 psi. Data acquisition and processing were performed with Agilent MassHunter Qualitative Analysis and Agilent MassHunter Quantitative Analysis.

Stock solutions of APX and internal standard (IS) were prepared in methanol at concentrations of 400 μg/mL and 111 μg/mL, respectively. Samples were prepared by adding 50 μL aliquot of APX solution into 50 μL of blank plasma, followed by adding 900 μL aliquot of IS solutions to obtain 1, 2, 5, 10, 20, 50, 100 ng/mL APX calibration standards and 1, 4, 40, 80 ng/mL for quality control (QC) samples, with a constant IS concentration of 100 ng/mL. After vortexing and centrifugation at 1500 rpm for 10 min, the supernatant was removed using a 0.2 μm PTFE syringe. A series of APX calibration, QC and IS solutions were prepared according to a serial dilution resulting into the concentrations as: 2000, 1000, 400, 200, 100, 40 and 20 ng/mL for APX calibration solutions; and 1600, 800, 80, 20 ng/mL for QC solutions and 11 ng/mL for IS solution. Calibration curves (linearity) with 7 calibration levels from 1 to 100 ng/mL with a double blank (no APX, no IS) and a zero blank (no APS, IS only) were constructed with the ratios of the peak area of APX to that of IS against APX concentration. The calibration curves were fitted using a simple linear regression model. 

Lower limit of quantification (LLOQ) and carryover was evaluated with 6 replicates.

Double blank and LLOQ samples in the rat plasma matrix from 6 individual sources were analyzed. The presence of any interference in double blank and LLOQ samples were carefully monitored. 

QC samples in one batch were analyzed in 5 replicates to establish intra-day accuracy and precision. Inter-day accuracy and precision were established by analyzing QC samples in triplicate. Relative error (%) and coefficient of variation (%) were calculated. 

Matrix effect was evaluated by analyzing 4 ng/mL and 100 ng/mL samples of the rat plasma matrix from 6 individual sources. Coefficient of variation (%) was calculated for each concentration.

Table 1 shows APX LC-MS/MS validation results.

#### 2.2.13. Statistical Analysis

Statistical analyses were carried out using the IBM SPSS Statistics (Statistical Product and Service Solutions (SPSS) software (version 25.0, IBM, New York, NY, USA). 

## 3. Results and Discussion

### 3.1. Preparation of Apixaban (APX) Solid Dispersions (SDs)

The saturation solubility study, the screening of solvents study was performed prior to the preparation of APX SDs. As shown in Table 2**,** the saturation solubility of APX in water was 0.031 mg/mL; meanwhile, methylene chloride, acetonitrile, methanol and acetone showed their capability of sufficiently solubilizing APX more than 30 times compared to that of water. Solubility of APX in chloroform was the highest at 13.87 mg/mL. However, due to intermolecular hydrogen bond formation, APX was re-precipitated quickly in chloroform. In this study, we chose acetone—a class 3 solvent with low toxic potential—as an adequate organic solvent to prepare SDs. 

In a broad range of polymers for SDs to choose from, we focused on the products which were reported to show a significant improvement in solubility for insoluble drugs [21]; additionally, carriers with P-glycoprotein inhibition effects are preferred because (1) normally an inhibitor of P-glycoprotein is co-administered with the drug to enhance drug absorption [22] and (2) APX elimination is heavily affected by the P-glycoprotein. Saturation solubility of APX in water with different hydrophilic carriers at various concentrations (0.1%, 0.5%, 1%) were also measured and summarized in Table 3. As the concentration of hydrophilic carrier increased, the saturation solubility of APX increased except for in the case of Eudragit^®^ L100. This can be explained by the presence of the amine group in APX’s chemical structure (Figure 1), and Eudragit^®^ L100 is known to form a matrix with weak basic drugs with high solubility in the stomach and decrease their solubility at high pH. As a pH-dependent polymer candidate, Eudragit^®^ L100 affects the solubility of poorly soluble drugs in SDs based on a pH-dependent mechanism. Among the screened carriers, Kollidon^®^ VA 64, Kollidon^®^, Soluplus^®^, Kolliphor^®^ P 407, Kolliphor^®^ P 188 and Kolliphor^®^ TPGS showed good effects on solubility; therefore, these carriers were selected as candidates for SD preparation. Moreover, among them, Soluplus^®^, Kolliphor^®^ P 407, Kolliphor^®^ P 188 and Kolliphor^®^ TPGS are the ones that exhibit a P-glycoprotein inhibition effect [22]. 

Firstly, APX SDs were prepared by the kneading method. APX was dispersed into a hydrophilic carrier (1:5 *w*/*w*) as the drug-to-PVP ratio was reported to be ideal at 1:5 after a series of dissolution testing of SD systems of a new lead compound in PVP K30 [23]. Table 4 shows the saturation solubility of the formed APX SDs by the kneading method in water. The highest solubility value belongs to Kolliphor^®^ TPGS APX SD, followed by Soluplus^®^ APX SD and Kollidon^®^ VA 64 APX SD. We then proceeded to use these three carriers for the solvent evaporation method. 

The solvent evaporation method, which is one of the most used methods in the pharmaceutical industry for the preparation of SDs [24], was applied to prepare the final APX SDs samples. Table 4 also shows the saturation solubility of the formed APX SDs by the solvent evaporation method in water. All SDs with different hydrophilic carriers and different APX-to-carrier ratios showed a higher increase in saturation solubility as compared to APX. This can be explained by the improvement of wetting of drug particles and localized solubilization by the hydrophilic carriers. SDs which are prepared by solvent evaporation methods are significantly more soluble compared to SDs which are prepared by the kneading method. Formulation ASD-4 exhibited the maximum saturation solubility with Soluplus^®^ at 1:5 ratio compared to ASD-1 and ASD-2 with Kolliphor^®^ TPGS and Kollidon^®^ VA 64 at the same APX-to-carrier ratio. Soluplus^®^ not only has an amphiphilic structure, but it also has a detectable critical micelle concentration of 7.6 mg/L [25], which is much lower compared to that of other classical low-molecular-weight surfactants. It seems that Soluplus^®^ formed micelles and maintained a high supersaturation of APX by inhibiting both APX nucleation and crystal growth. In the case of Soluplus^®^, as the optimized drug-to-Soluplus^®^ ratio was previously reported as low as 1:2 [26,27], herein we confirmed that the saturation solubility of APX increased as more Soluplus^®^ was included in the system [28]. ASD-3, ASD-4 and ASD-5 were further evaluated.

Drug content values of ASD-3, ASD-4 and ASD-5 were measured by high-performance liquid chromatography (HPLC). Obtained values were between 100 ± 5% of the nominal values.

### 3.2. Differential Scanning Calorimetry (DSC) Analysis

The DSC thermograms of APX, Soluplus^®^, physical mixture (PM) of APX and Soluplus^®^ (1:5), and SDs (ASD-3, ASD-4, and ASD-5) are shown in Figure 2. Pure APX showed a sharp endothermic peak at 238 °C which is the melting point of the drug (Figure 2A). The thermogram of PM corresponded to the summation of crystalline APX and Soluplus^®^ thermogram, suggesting no interaction between the two species. However, ASD-3, ASD-4 and ASD-5 showed the absence of sharp endothermic peaks, indicating the existence of an interaction between crystalline APX and Soluplus^®^ (Figure 2B). Interestingly, there was a remaining small peak at around 238 °C in the thermogram of ASD-3, which means the crystallinity of APX in ASD-3 was drastically decreased in Soluplus^®^, but the amorphous degree of APX in ASD-3 was less than that of ASD-4 and ASD-5, and ASD-5 exhibited the highest amorphous degree. The crystalline was confirmed for ASD-3, but not for ASD-4 and ASD-5.

### 3.3. Fourier Transform Infrared Spectroscopy (FT-IR)

FT-IR patterns were detected for APX, Soluplus^®^, PM and formulations ASD-3, ASD-4 and ASD-5. As shown in Figure 3, the FTIR spectrum obtained for the neat amorphous APX showed several significant differences. The FT-IR spectrum of APX showed bands at 3309 cm^−1^, 3482 cm^−1^ for O-H/N-H stretching, 2910, 2867 cm^−1^ for C-H stretch, 1679, 1627 cm^−1^ N-C and -C=O stretching vibration for amide, 1593 cm^−1^ for C-H bending, 1293, 1254 cm^−1^ for C-O ether stretch and 1144, 1038 cm^−1^ for C-O/C-N stretch. The FT-IR spectra for Soluplus^®^ showed major bands at 2923, 2857 cm^−1^ for C-H stretch, 1731 cm^−1^ for OC(O)CH3, 1631 cm^−1^ for N-C and -C=O stretching vibration for amide and 1230 cm^−1^ for C-O ether stretch. The spectrum of PM showed major bands of both APX (3482, 3308 cm^−1^ for O-H/N-H stretching) and Soluplus^®^ (2928, 2910 cm^−1^ for C-H stretch). However, the spectrum of ASD-3, -4, -5 showed absence in the peak position of O-H/N-H stretching of APX at 3500–3200 cm^−1^ and decreased peak intensity of C=O stretching of Soluplus^®^ at 1760–1630 cm^−1^ was observed. These shifts in ASD-3, -4 and -5 confirmed formation of potential hydrogen bonding between the APX hydroxy/amine group (H donor) and Soluplus^®^ ester C=O group and internal amide C=O group.

### 3.4. Powder X-ray Diffraction (PXRD)

PXRD patterns were detected for APX, Soluplus^®^, PM and formulations ASD-3, ASD-4 and ASD-5. As shown in Figure 4, APX has distinct peaks at 12.78°, 13.84°, 16.98°, 18.38° and 22.1°, which provide a clear indication that APX is in crystalline form. On the other hand, Soluplus^®^ showed a broad peak which suggests it is in its total amorphous state. The PXRD pattern of PM showed a complex of both APX and Soluplus^®^ with less intense but still sharp peaks corresponding to that of APX crystalline, and no observable distinct peaks of new crystals of a new complex/compound. It was also observed that the intensity of peaks of APX in APX SDs were found to be markedly reduced and not sharp when compared to that of the pure APX and PM. It is suggested that the crystallinity of APX was reduced in SDs as compared to APX and PM.

### 3.5. In Vitro Dissolution Studies

The dissolution behavior of APX and APX SDs was studied by using disc intrinsic dissolution rate (DIDR) and powder dissolution. 

DIDRs were determined for the compressed discs of SDs. The plots of the dissolved amount of APX (mg/cm^2^) versus time for pure APX, ASD-3, ASD-4 and ASD-5 are shown in Figure 5. All the formulations showed perfect linearity with a correlation coefficient value over 0.99. It was clear that SDs successfully increased the dissolution rate of APX compared to crystalline APX. The DIDRs were determined as follows: ASD-4 > ASD-3 > ASD-5 > PM > APX, where the values are 0.024, 0.017, 0.014, 0.012 and 0.005 (mg/min/cm^2^), respectively. According to an earlier report from this DIDR study result, APX can be classified as ‘low solubility’ as the DIDR of APX is lower than 0.1 mg/min/cm^2^ [9]. This discrepancy, which occurs between APX BCS classification and APX DIDR, should be noted as dose is considered according to the drug BCS group, while DIDR does not consider the effect of dose. The DIDR result is the evidence that APX should be solubilized although it is classified as BCS class III.

The powder dissolution rate of pure APX and SDs were evaluated in sodium phosphate buffer with 0.05% SLS under sink condition. As shown in Figure 6, it was found that all SDs showed a significantly higher dissolution profile compared to that of pure APX. In agreement with the DIDR test result, the dissolution rate from the powder dissolution study was determined as follows: ASD-4 > ASD-3 > ASD-5 > PM > APX. During the first 30 min, the cumulative dissolved APX was almost 80% for the ASD-3 and it reached its steady state before 120 min with a complete release of APX. However, the cumulative dissolved APX was only 40% for pure APX at the end of 180 min, indicating that the prepared SDs dramatically increased the dissolution rate of crystalline APX. 

Both the DIDR study and powder dissolution study showed that the dissolution rate increased with the higher amount of Soluplus^®^ (ASD-4 > ASD-3), but the dissolution rate decreased when the amount of carriers was going beyond an acceptable limit in the dispersed system (ASD-4 > ASD-3 > ASD-5). This may be because the crystalline APX did not fully transform into an amorphous state in the case of ASD-3, leading it to need more energy to be dissolved. In the case of ASD-5, which showed the lowest dissolution rate among SDs, we think that Soluplus^®^ formed a barrier outside the surface of SD particles caused by its viscous properties.

### 3.6. Ex Vivo Permeation Test

Permeabilities of pure APX and ASD-4 across the rat jejunum were assessed using the modified Ussing chamber method. The P_app_ was calculated every hour using Equation (2). The permeability enhancement ratio was also calculated by dividing the P_app_ of ASD-4 by that of pure APX at the same time point. The result of the ex vivo permeation test is summarized in Table 5.

The pure APX showed poor permeability as expected and the calculated P_app_ for three hours were 9.79, 15.60 and 20.26, respectively. On the other hand, the P_app_ of ASD-4 were 24.88, 18.03 and 17.86. Although it showed gradual decrease over time, P_app_ of ASD-4 across the intestinal membrane were increased compared to those of pure APX except for 3 h. During the first 1 h, the permeability of ASD-4 was increased 2.54 times more than that of raw APX. That backs up the hypothesis that Soluplus^®^ improves the intestinal permeability of APX by inhibiting P-glycoprotein.

### 3.7. In Vivo Pharmacokinetic Study 

The plasma pharmacokinetic profiles of APX following an oral administration of raw APX and ASD-4 at the drug dose of 2 mg/kg ar shown in Figure 7 and Pharmacokinetic parameters are summarized in Table 6. As shown in Figure 7, ASD-4 increased the AUC and T_max_ by 2.31 times and 2.29 times, respectively, than raw APX. In addition, the plasma concentration of ASD-4 decreased slower than raw APX after 2 h. However, C_max_ of ASD-4 decreased 0.91 times compared to raw APX. These results showed an improvement in oral bioavailability despite a slight decrease in C_max_. Because of the P-glycoprotein inhibition effect of Soluplus^®^ in ASD-4, the plasma concentration increased after 2 h with an increase in the absorption rate of APX in the small intestine area. These results are consistent with observations from ex vivo permeation studies.

## 4. Conclusions

Solid dispersion (SD) is a well-known technique that helps to improve the solubility of poorly soluble drugs. Apixaban (APX) SD formulations with hydrophilic carriers were prepared by the solvent evaporation method. Depending on the interaction between APX and the studied hydrophilic carriers, the solubility of resulting APX SD varied. The carrier showing the highest solubility improvement was Soluplus^®^. The optimal composition of APX SD was APX: Soluplus^®^ of 1:5 (ASD-4) and it effectively improved the dissolution rate, ex vivo permeability, and oral administration bioavailability in rats compared to with raw APX. Although further studies such as cytotoxicity studies should be conducted for APX SDs, novel APX modified-release formulations and routes of administration employing ASD-4 can be developed. We expect that ASD-4 products increase the bioavailability of APX and reduce irritation for patients. 

## Figures and Tables

**Figure 1 pharmaceutics-15-00907-f001:**
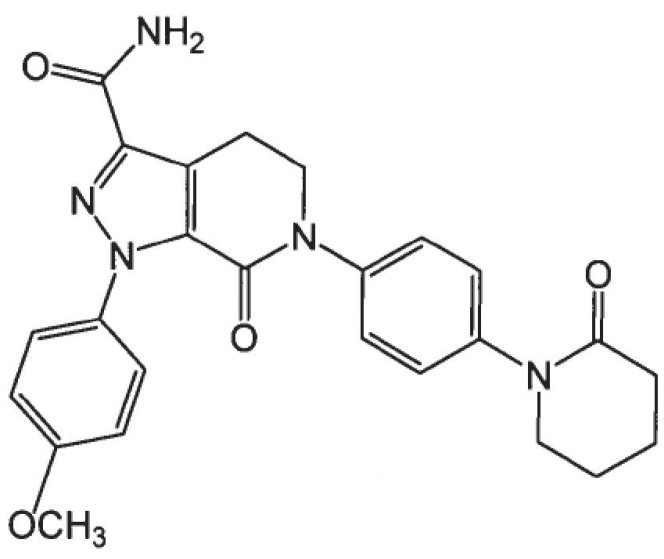
Chemical structure of APX.

**Figure 2 pharmaceutics-15-00907-f002:**
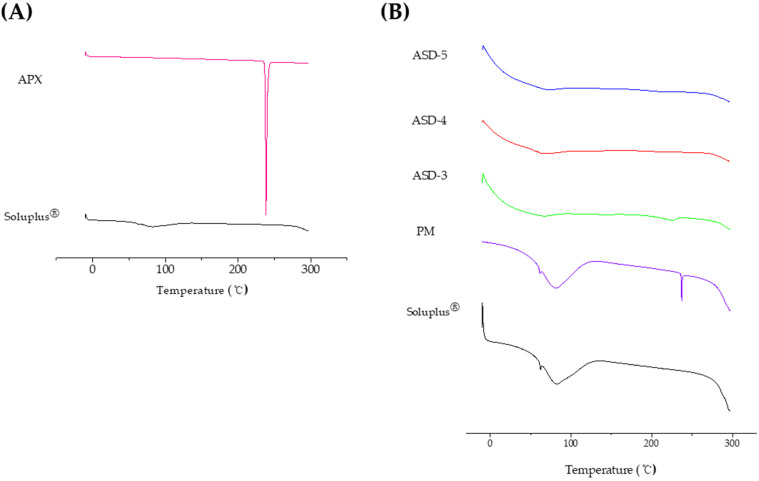
DSC thermograms of (**A**) pure APX and Soluplus^®^ (**B**) APX-Soluplus^®^ PM and Formulations ASD-3, ASD-4 and ASD-5.

**Figure 3 pharmaceutics-15-00907-f003:**
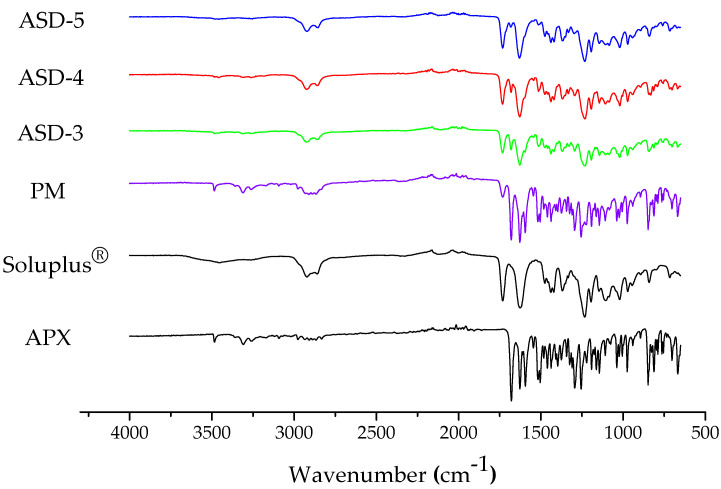
FT-IR spectra of APX, Soluplus^®^, PM, and ASD-3, ASD-4, and ASD-5.

**Figure 4 pharmaceutics-15-00907-f004:**
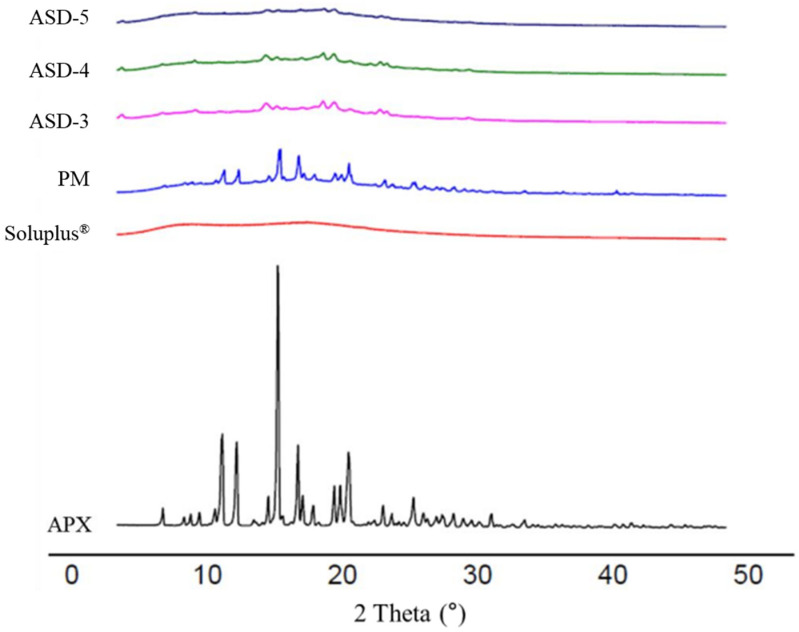
PXRD pattern of APX, Soluplus^®^, PM, and ASD-3, -4, -5.

**Figure 5 pharmaceutics-15-00907-f005:**
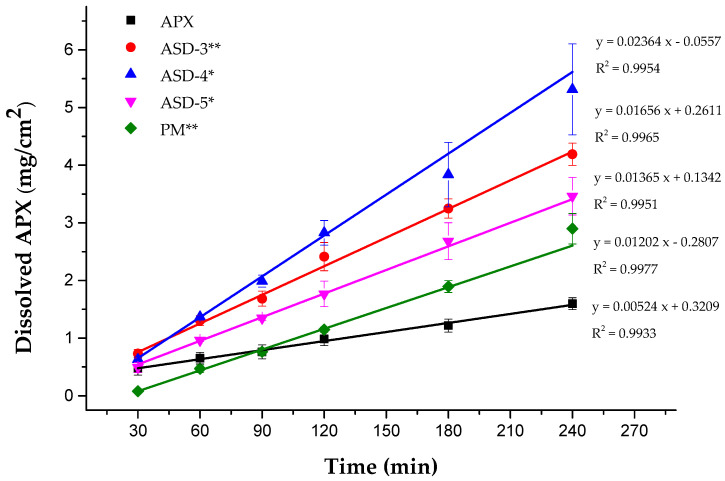
DIDR profiles of APX, ASD-3, ASD-4, ASD-5 and PM. * *p* < 0.05, ** *p* < 0.01.

**Figure 6 pharmaceutics-15-00907-f006:**
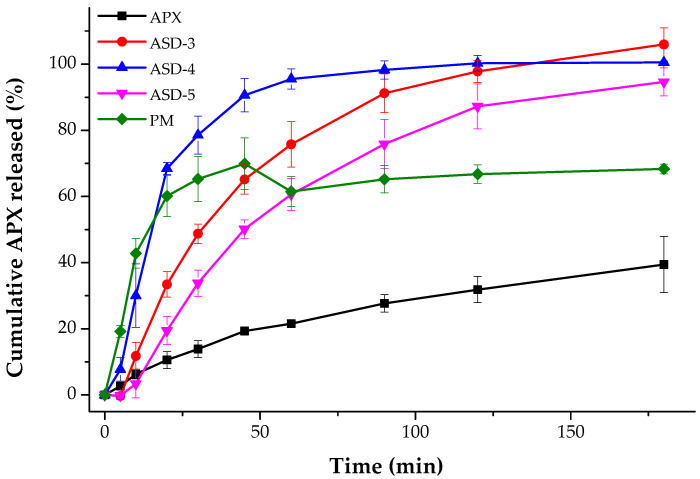
Powder dissolution rate of APX, ASD-3, ASD-4, ASD-5 and PM.

**Figure 7 pharmaceutics-15-00907-f007:**
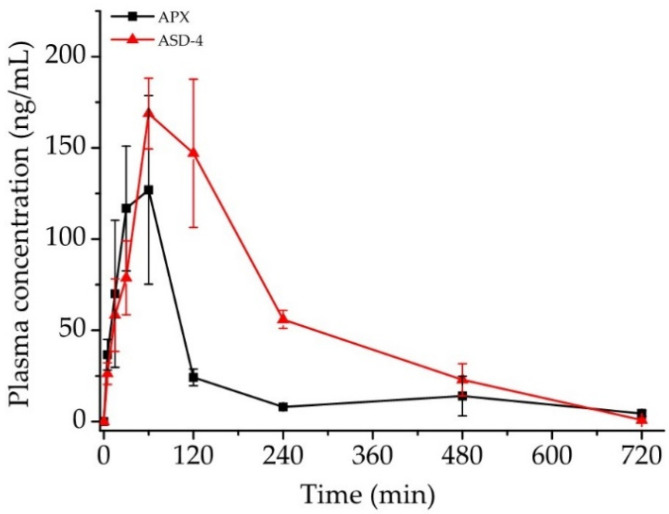
Pharmacokinetic profile of oral APX and ASD-4 in rats (mean ± S.D, n = 3). The dose was equivalent to 2 mg/kg of apixaban.

**Table 1 pharmaceutics-15-00907-t001:** Summary of LC-MS/MS validation results.

Parameter	Result
Precursor Ion *m*/*z*	460.2 [M + H] ^+^
Product Ions *m*/*z*	443.02
Range	1 to 100 ng/mL
Selectivity	No interferences
Carry-over	Not observed
Calibration curve	Y = 0.0164x − 0.0027, R^2^ = 0.9997
Accuracy and Precision	Concentrations: 1, 40, 40 and 80 ng/mL Intra-batch RE *: −18.1%, −1.9%, 7.7%, 12.5% Inter-batch RE: 6.3%, 10.4%, −0.31%, −0.83% Intra-batch CV **: 9.5%, 2.3%, 3.5%, 5.5% Inter-batch CV: 3.6%, 8.9%, 4.8%, 4.0%
LLOQ	1 ng/mL, S/N ratio = 94.63
Matrix effect	Low concentration CV: 9.3% High concentration CV: 6.0%

* RE: relative error, ** CV: Coefficient of variation.

**Table 2 pharmaceutics-15-00907-t002:** Saturation solubility of APX in organic solvent.

Solvent	Saturation Solubility (mg/mL)
Chloroform	13.87
Water	0.031
Ethyl acetate	0.06
2-Propanol	0.13
Ethanol	0.43
Acetone	1.01
Methanol	2.21
Acetonitrile (ACN)	2.38

**Table 3 pharmaceutics-15-00907-t003:** Saturation solubility of APX in water with different hydrophilic carriers at 0.1%, 0.5% and 1%.

Hydrophilic Carrier		Saturation Solubility (µg/mL)		
Product	Description	0.1%	0.5%	1%
Eudragit^®^ L100 *	Methacrylic acid-methyl methacrylate (1:1)	34.23	12.89	3.73
Eudragit^®^ S100 *	Methacrylic acid-methyl methacrylate (1:2)	31.63	32.89	30.69
Eudragit^®^ E PO	Butyl methyl methacrylate, dimethyl aminoethyl methacrylate, methacrylate, methyl methacrylate	33.11	33.97	34.85
Kollidon^®^ VA 64	Povidone, vinylpyrrolidone-vinyl acetate	38.75	46.37	57.42
Kollidon^®^	Povidone (PVP)	35.53	41.46	43.67
	Poly (vinyl alcohol) (PVA)	34.89	36.94	38.60
Soluplus^®^ *	Polyvinyl caprolactampolyvinyl acetate-polyethylene glycol	41.01	59.61	53.61
Kolliphor^®^ P 407 *	Poloxamer 407	38.68	43.30	44.49
Kolliphor^®^ P 188 *	Poloxamer 188	36.62	42.55	43.72
Kolliphor^®^ TPGS *	D-alpha tocopheryl polyethylene glycol 1000 succinate	39.63	58.98	80.35

* Hydrophilic carrier with p-glycoprotein inhibition effect [22].

**Table 4 pharmaceutics-15-00907-t004:** Saturation solubility of SDs prepared by kneading method.

Preparation Method	Hydrophilic Carrier (APX-to-Carrier Ratio *w*/*w*)		Saturation Solubility (µg/mL)
Kneading method	Kollidon^®^ (1:5)		40.83
	Kolliphor^®^ P 188 (1:5)		46.78
	Kolliphor^®^ P 407 (1:5)		48.78
	Kollidon^®^ VA 64 (1:5)		55.66
	Soluplus^®^ (1:5)		57.83
	Kolliphor^®^ TPGS (1:5)		69.52
Solvent evaporation method	Formulation	Hydrophilic carrier (APX-to-carrier ratio *w*/*w*)	Saturation solubility (µg/mL)
	ASD-1	Kolliphor^®^ TPGS (1:5)	85.7
	ASD-2	Kollidon^®^ VA 64 (1:5)	95.8
	ASD-3	Soluplus^®^ (1:3)	131.9
	ASD-4	Soluplus^®^ (1:5)	183.2
	ASD-5	Soluplus^®^ (1:10)	200.9

**Table 5 pharmaceutics-15-00907-t005:** Apparent permeability coefficient (P_app_) of pure APX and SDs.

Donor Solution	Apparent Permeability Coefficient (×10^−6^ cm/s)		
	0–1 h	0–2 h	0–3 h
APX	9.78	15.60	20.26
ASD-4	24.88	18.03	17.86
Enhancement ratio	2.54	1.16	0.88

**Table 6 pharmaceutics-15-00907-t006:** Pharmacokinetic parameters of oral APX and ASD-4 in rats (mean ± S.D, n = 3).

Parameter	APX	ASD-4
Cmax (ng/mL), Tmax (min)	Rat 1, 163.01, 15 Rat 2, 240.77, 60 Rat 3, 184.30, 30	Rat 1, 249.81,120 Rat 2, 134.02, 60 Rat 3, 161.76, 60
Mean C_max_ (ng/mL)	196.0 ± 40.18	178.9 ± 55.41
Mean T_max_ (h)	0.58 ± 0.38	1.33 ± 0.58
AUC_0→12_ (ng∙h/mL)	16,962.6 ± 3601.48	39,192.1 ± 11,580.14

## Data Availability

Not applicable.

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
