# Peer review of "Preparation of Apixaban Solid Dispersion for the Enhancement of Apixaban Solubility and Permeability"

_pharmaceutics, 2023, doi:10.3390/pharmaceutics15030907_

Round 1
Author Response
Dear Reviewers,
Please see the attachment.
Enclosed please find the revised manuscript entitled “Preparation of Apixaban Solid Dispersion for the Enhancement of Apixaban Solubility and Permeability
(Manuscript ID: pharmaceutics-2209890)” for consideration as an article in
Pharmaceutics.
We are grateful for the thoughtful comments of the reviewers, whose contribution to the clarity and accuracy of the manuscript is substantial. Below is a point-by-point description of how we have modified the manuscript according to the reviewer’s comments or otherwise answered the reviewer's questions. On almost all points, we have been able to modify the manuscript exactly as suggested by the reviewers.
Our responses are highlighted in Blue in this letter. And the corresponding revisions in the body of the manuscript are highlighted in Red.

Reviewer 2 Report
Dear Authors, The manuscript titled Preparation of Apixaban Solid Dispersion for the Enhancement of Apixaban Solubility and Permeability treats the using of hydrophilic carriers to prepare solid dispersion of selected model drug by solid dispersion.
After some minor corrections I recommend this manuscript to be published in pharmaceutics.
Line 38: According to the USP/NS the Authors describe Apixaban as active substance that belongs to BCS class III (high solubility and low permeability), please explain such phenomenon because the Authors described Apixaban as practically insoluble.
Line 44: In the introduction section please add information about any previous studies regarding Apixaban and solid dispersion. Please add briefly information about carriers used, preparation method, etc.
Line 61: Please explain shortly why Rivaroxaban is used.
Line 112: Please add weight ratio of the substances used during preliminary studies. What carriers were tested?
Line 125: Please add the description of the analysis of drug content.
Line 167: It should be medium instead of medum. Why such a medium was chosen to performed dissolution studies?
Line 168: What was time intervals? Please add detailed information.
Line 313: Please add detail information which data corresponds to Figure 1 panel A and Figure 1 panel B.
Author Response
Dear Reviewers,
Please see the attachement.
Enclosed please find the revised manuscript entitled “Preparation of Apixaban Solid Dispersion for the Enhancement of Apixaban Solubility and Permeability
(Manuscript ID: pharmaceutics-2209890)” for consideration as an article in
Pharmaceutics.
We are grateful for the thoughtful comments of the reviewers, whose contribution to the clarity and accuracy of the manuscript is substantial. Below is a point-by-point description of how we have modified the manuscript according to the reviewer’s comments or otherwise answered the reviewer's questions. On almost all points, we have been able to modify the manuscript exactly as suggested by the reviewers.
Our responses are highlighted in Blue in this letter. And the corresponding revisions in the body of the manuscript are highlighted in Red.

Reviewer 3 Report
The study entitled: “Preparation of apixaban solid dispersion for the enhancement of apixaban solubility and permeability” describes the preparation and characterization of solid dispersions of an anticoagulation drug (Apixaban) in Soluplus®, among other carriers, as well as bioavailability and permeability measurements.
In order for the article to be of sufficient quality for publication, the following considerations must be taken into account by authors.
1. It is highlighted in the introduction (lines 41-43): “To the best of our knowledge, there is no research done to improve APX solubility and permeability, ultimately to enhance its oral bioavailability” and this is not true and requires further literature review by the authors.
There are several articles focused on the study of the improvement of the solubility of this active ingredient, including, for example:
- Patent CN 108096205, 2018. Zhoye Y.Y.
In this patent, apixaban tablets are formed employing apixaban solid dispersions with mannitol as carrier and auxiliary components (microcrystalline cellulose, PVP and silica power).
- “Solubility Enhancement of BCS classified II/IV drug-solid dispersion of Apixaban by solvent evaporation”. Asati et al., 2020. International Journal of Pharmaceutical Investigation, 2020, 10(4): 430-436
In this work, solid dispersions with hydrophilic polymer (HPMC, PEG 600 and PVP K30) are prepared by spray drying method showing improvement in aqueous solution.
- “Cocrystal of Apixaban-Quercetin: improving solubility and bioavailability of drug combination of two poorly soluble drugs”. Zhang et al, 2021. Molecules, 26,2677
The cocrystal formulation with quercetin has been proved as an adequate strategy to improve the solubility and bioavailability characteristics of apixaban.
2. Why is it said that 1:5 w/w kneading samples can be considered as the ideal drug-to-polymer ratio in this study whereas 1:3 ad 1:10 w/w are also studied? (lines 283-285). This choose requires a further explanation.
3. With respect to DSC analysis, I do not agree with the following sentence: “However, ASD3, ASD4 and ASD 5 showed the absence of sharp endothermic peaks indicating crystalline APX in Soluplus® was forced to enter its amorphous state”.
The interaction drug-polymer can be the main reason of the lack of drug crystallinity.
In addition, the thermal effect around 150°C in 1:3 and 1:5 w/w solid dispersions must be mentioned and explained.
4. In XRD diffraction patterns of solid dispersions, some of the reflections corresponding to drug can be detected with lower intensity as result of the interaction with the polymer but not a complete amorphous state as mentioned. (lines 333-334).
5. In dissolution studies, it must be indicated if DIDRs values (line 346) of ASD3, 4 and 5 (0.024, 0.017 and 0.014 mg/min/cm2) presents significant differences.
6. In the conclusion it is said that “the optimal composition of APXSD was APX:Soluplus® of 1:5 (ASD 4)”. Would be it possible to prepare a formulation with such a high polymer amount?
7. Recommendation: Citotoxicity studies could improve the quality of work

Author Response

(The authors gave the same response as above.)

Reviewer 4 Report
This article reports that formation of solid dispersion for Apixaban to enhance its oral absorption through solubility and permeability. The formulation resulted in improvement of solubility and permeability in vitro test, in addition oral absorption of rat was also confirmed. These are worth for publication as pharmaceutical research. However, the authors need to revise significantly the manuscript before publication because some issues were confirmed. The comments are listed here, and please refer to them.
“Abstract”
1. The authors refer to FTIR as an analytical method for solid dispersion (line 17), but the results were not shown in the main text? Why is this? Spectroscopic analysis is important method to analyze molecular state such as intermolecular interaction. Please add it into the text.
“1. Introduction”
2. Apixaban was written as lower soluble drug in this article, but biopharmaceutical classification was Class III corresponding to higher solubility and lower permeability (line 39). What is author’s consideration? If the author would define this drug as class VI, please write it in Introduction based on further references such as definition of BCS.
3. Introduction is too short. Please add information about discussion points of this article such as definition of solid dispersion and issue of them (preparation method, stability, type of polymeric carrier et al.).
“2. Material and Methods”
4. Linefeed is not necessary for each material (line 60-90).
“3. Results and discussion”
5. What means “crystallization in chloroform”. The other solvents were also used for saturation solubility for crystal compound. The authors would mean solvation with chloroform? (Table 1).
6. The full name is needed for the abbreviation “DW” (line 103).
7. The different effect of the polymers on the solubility of the API is shown in Table 2; Eudragit L100 decreased the solubility of the API with is concentration. The authors need to discuss the reason for these phenomena. Eudragit L is a pH-dependent polymer which might affect the solubility in water. Please add this information with the chemical structure of the API. This information should help readers to recognize enhancement of solubility mechanism based on the functional groups and interaction with the polymer.
8. The result of kneading method is not important because this test was done using the suspended samples (Table 3).
9. The result of the solvent evaporation showed that Soluplus gave highest solubility of the API. The authors discussed the physicochemical properties of Soluplus resulted in good effect. However, no evaluation was performed for the precipitated samples after the solvent evaporation. The authors need to measure them by the XRPD and DSC et al. to discuss the difference of the solubility.
10. In the discussion of the DSC result, the absence of sharp peak corresponds to amorphization. However, the broadened endothermic peak was confirmed in the profile of ASD-3. The ASD-4 also showed small peak around 160-170 Cel. The XRDP patterns also support these results; the crystalline peaks appear in the profiles of ASD-3, ASD-4, and ASD-5 (Figure 2). This indicates that the solid dispersion used in this study was not amorphous state. This is a critical point for discussion of the data. The solid dispersion is commonly used as amorphous formulation; hence it is needed that the crystalline dispersion is represented in the title and text. I recommend the authors to try preparation of amorphous solid dispersion and evaluation of them.
11. “It is suggested that APX in SDs was in complete 334 amorphous state as compared to APX and PM.” was written in line 334-335. As mentioned above, the crystalline peaks were confirmed for the evaporation samples, suggesting that crystalline remained. The author need to try preparation of complete amorphization for discussion as the solid dispersion.
12. Please add the dissolution results of PM in Figure 3 and 4.
13. Why permeability of the ASD gradually decreased? Please discuss it (line 381-382).
14. Are pharmacokinetic parameters, right? (Table 5). For example, Cmax of API and ASD-4 are 196.0 and 178.9, respectively. However, Figure 5 shows higher maximum concentration of ASD-4.
That’s all.
Author Response

(The authors gave the same response as above.)

Reviewer 5 Report
Manuscript pharmaceutics-2209890
Title: Preparation of Apixaban Solid Dispersion for the Enhancement of Apixaban Solubility and Permeability
In this paper the authors use the concept of Solid Dispersions to develop a powder product of Apixaban that would make improvement of the solubility and permability primarily by assisting the wetting of the drug. Solid dispersion method was compared with wet mixing of the drug with the same hydrophilic polymers. The work is interesting and well documented by ex-vivo and in-vivo experiments. The low dose of the proposed drug is not a negative factor for conducting the research as solubility and permeability improvement is always welcome
Some points that may be consider for paper improvement are given below
Acetone is flammable and generally avoided as a bulk solvent. Why did the authors chose this solvent?
Authors decided for acetone on the basis of drug solubility in acetone but did not explain the reason for the use of this solvent low polarity solvent toogether with hydrophilic polyemers included in the solid dispersions
Authors should organize the comparative experiments between the two preparation methods for the solid dispersion, e.g by kneading and solvent evaporation in a statistical manner so that firm conclusions can be derived.
Other points
Line 56, 57 .
The objective of the study mentioned 'to make patients feel more comfortable appears to be out of the scope of the study
Line 103 - please define 'DW'
Line 128 - please correct to 'Differential ....'
Line 145 - Experimental description is incompletee. What part of tablet was exposed to liquid? how was it fixed?
Line 166 - gelatin capsules?
Line 284 - What is the similarity of PVP with the experimental polymers used in this study?
Author Response

(The authors gave the same response as above.)

Round 2
Reviewer 3 Report
The quality of study has been improved
Author Response
Reviewer 1:
The quality of study has been improved
We appreciate your comment.
Reviewer 4 Report
I am pleased to be invited as a reviewer for Pharmaceutics. The revised version of the manuscript pharmaceutics- 2209890 “Preparation of Apixaban Solid Dispersion for the Enhancement of Apixaban Solubility and Permeability” was reviewed. The authors revised the manuscript according to the reviewer’s comment. Most of the points responded with the comment to the reviewer. However, only the result of DSC should be re-revised. The authors re-tried the preparation of the solid dispersion and mentioned that ASD-3, ASD-4, and ASD-5 showed absence of the melting peak corresponding to amorphization. But ASD-3 gave broadened endothermic peak around 200-250 °C. Please replace the result and discussion to like “the crystalline was confirmed for ASD-3, but not for ASD-4 and ASD-5”. This result agrees with the increment of the solubility depending on the amount of Soluplus, which should improve the quality of this article.
Author Response
Dear reviwer,
Please see the attachment.
